# A fast comparative genome browser for diverse bacteria and archaea

**Morgan N. Price** [ID]*, **Adam P. Arkin**

Environmental Genomics and Systems Biology, Lawrence Berkeley National Lab, Berkeley, California, United States of America

* funwithwords26@gmail.com

## Abstract

Genome sequencing has revealed an incredible diversity of bacteria and archaea, but there are no fast and convenient tools for browsing across these genomes. It is cumbersome to view the prevalence of homologs for a protein of interest, or the gene neighborhoods of those homologs, across the diversity of the prokaryotes. We developed a web-based tool, *fast.genomics*, that uses two strategies to support fast browsing across the diversity of prokaryotes. First, the database of genomes is split up. The main database contains one representative from each of the 6,377 genera that have a high-quality genome, and additional databases for each taxonomic order contain up to 10 representatives of each species. Second, homologs of proteins of interest are identified quickly by using accelerated searches, usually in a few seconds. Once homologs are identified, *fast.genomics* can quickly show their prevalence across taxa, view their neighboring genes, or compare the prevalence of two different proteins. *Fast.genomics* is available at https://fast.genomics.lbl.gov.

**Data Availability Statement:** Fast.genomics is available at http://fast.genomics.lbl.gov. The source code and the database have also been archived at figshare (https://doi.org/10.6084/m9.figshare.24010353.v1).

## Introduction

As of July 2023, the Genome Taxonomy Database lists about 85,000 species of bacteria and archaea and about 20,000 genera [1]. Although many of these taxa are known only from low-quality metagenome-assembled genomes, high-quality assemblies [2, 3] are available for over 6,000 genera and over 29,000 species. These genomes contain valuable information about the functions of genes and their ecological roles. For example, knowing which other protein families are encoded near a protein of interest, across diverse prokaryotes, can allow one to propose the protein's function [4–6]. If homologs of two proteins co-occur, this suggests a close functional relationship [6, 7]. And the distribution of a protein family across taxa can be used to identify horizontal gene transfer events [8] or to give hints as to the protein's ecological role [9].

All of these analyses depend on identifying homologs of the protein(s) of interest across many genomes. The standard approach is to use protein BLAST [10] or a similar tool to find homologs, but given the size of current genome databases, this typically takes over 10 minutes, which precludes interactive analysis. An alternative is to pre-compute groups of homologous proteins, using resources such as PFam [11], TIGRFam [12], or eggNOG [13]. Examples of

**Funding:** This material by ENIGMA- Ecosystems and Networks Integrated with Genes and Molecular Assemblies (http://enigma.lbl.gov), a Science Focus Area Program at Lawrence Berkeley National Laboratory is based upon work supported by the U.S. Department of Energy, Office of Science, Office of Biological & Environmental Research under contract number DE-AC02-05CH11231 The funders had no role in study design, data collection and analysis, decision to publish, or preparation of the manuscript.

**Competing interests:** The authors have declared that no competing interests exist.

fast tools that rely on pre-computed homology groups include GeCoViz, for identifying genes that are conserved near a query gene [14]; AnnoTree, for viewing the taxonomic distribution of protein families [15]; and PhyloCorrelate, for identifying protein families that have a similar distribution as a protein family of interest [16]. However, these tools will not give useful results, or might give misleading results, if the protein family of interest is a poor match to the pre-computed families. Tools that rely on pre-computed homology groups can also be cumbersome to maintain due to the computational cost of comparing new genomes to the existing protein families.

To support interactive browsing of protein families across the diversity of bacteria and archaea, we built a new website, *fast.genomics* (https://fast.genomics.lbl.gov/). The key challenge for interactive browsing is the fast identification of homologs for a protein of interest. We selected one high-quality representative genome for each genus, so the main database of *fast. genomics* contains proteins from "only" 6,377 genomes. Because the Genome Tree Database (GTDB) aims for a roughly consistent age for each genus, as estimated using relative evolutionary divergence [1], these representatives should evenly cover the sequenced diversity of bacteria and archaea. In case searching the main database does not find enough homologs, *fast.genomics* also has a database for every taxonomic order, with up to 10 representatives of each species. Searching in the order that a protein belongs to will usually yield many additional homologs.

To speed up homology searches against the main database, *fast.genomics* uses MMseqs2 [17], an accelerated alternative to protein BLAST, and stores the MMseqs2 index in memory. Because MMseqs2 is optimized for large-scale searching, it does not benefit from multiple CPUs when handling a single query. To get around this, *fast.genomics* splits the candidates from the prefilter step of MMseqs2 into 10 lists and analyzes them in parallel. Parallel MMseqs2 takes an average of just 3.3 seconds per query, and is almost as sensitive as protein BLAST. Because some of the taxonomic orders contain thousands of genomes, *fast.genomics* also uses an accelerated strategy, based on clustering similar sequences, to search the order-level databases.

Once homologs have been identified, *fast.genomics* includes tools to view their gene neighborhoods, to view their taxonomic distribution, and to compare the distribution of the homologs of two proteins (also known as phylogenetic profiling or co-occurrence analysis).

## Results and discussion

### The *fast.genomics* databases

To ensure that the presence or absence of a gene family can be determined reliably, *fast.genomics* includes only high-quality genomes. Specifically, we required that an assembly be at least 90% complete and have at most 5% contamination, as assessed by CheckM [3], and that the assembly not be chimeric, as assessed by GUNC [18]. For metagenome-assembled genomes or single-cell amplified genomes, we required that they meet the MIMAG guidelines for a high-quality draft, [2], which means that they contain ribosomal rRNAs and most tRNAs. Using the April 2023 release of the Genome Taxonomy database, we identified 6,377 genera and 29,413 species with high-quality genomes (Table 1). All taxonomic assignments in *fast.genomics*, including genus and species names, are taken from GTDB. Roughly speaking, GTDB defines species as clusters of genomes with above 95% average nucleotide identity, and defines groups at higher taxonomic levels to have consistent ages [1].

### Overview of the *fast.genomics* website

The main function of *fast.genomics* is to select a protein or gene of interest and then to find its homologs. The analyst can select a protein using locus tags or protein identifiers from *fast.*

**Table 1. Statistics for the main database, for the biggest order-level databases, and for all databases combined.** The number of proteins in each database is the number of distinct sequences, not the number of protein-coding genes.

| Database | genomes | genera | species | proteins, millions | clusters, millions |
|---|---|---|---|---|---|
| Main | 6,377 | 6,377 | 6,377 | 21.8 | - |
| Pseudomonadales | 4,321 | 163 | 1,807 | 11.7 | 1.9 |
| Burkholderiales | 3,772 | 340 | 1,997 | 12.3 | 2.8 |
| Actinomycetales | 3,339 | 245 | 1,891 | 7.7 | 2.2 |
| Rhizobiales | 3,271 | 215 | 1,643 | 12.5 | 2.4 |
| Lactobacillales | 3,223 | 98 | 1,071 | 3.7 | 0.9 |
| Combined | 56,186 | 6,377 | 29,413 | 217.3 | 44.3 |

*genomics* as well as identifiers from a variety of other databases, including UniProt, the NCBI Protein database, RefSeq, or the Protein Data Bank. Or the analyst can enter a protein sequence. *Fast.genomics* also supports combined genus/annotation searches, such as "Escherichia thymidylate synthase". Finally, the analyst can search for a taxon of interest and view genomes within a given genus or species. From the genome page, the analyst can find a gene or protein or interest by searching the text annotations or by using Curated BLAST to find proteins that are similar to characterized proteins that have that annotation [19].

Once the protein sequence of interest has been specified, *fast.genomics* provides accelerated sequence searches, against either the main database or against an order-level database, followed by several ways of visualizing the results.

## Speed and sensitivity of finding homologs in the main database

To find homologs in the main database, *fast.genomics* uses fast parallel MMseqs2. To test the speed and sensitivity of our approach, we selected 1,000 proteins at random from the main database. We searched for homologs of these proteins in the main database using protein BLAST from NCBI's BLAST+ package [20], fast parallel MMseqs2, or MMseqs2 with the most sensitive settings. We ran each query separately, as would occur during interactive use. BLAST + and parallel MMseqs2 used 10 threads, and for MMseqs2, the index was loaded into memory beforehand.

We searched for up to 6,377 homologs with $E \leq 10^{-3}$ (6,377 is the number of genomes). These are the same settings used by the *fast.genomics* website. Fast parallel MMseqs2 took an average of 3.3 seconds per query, which was 7 times faster than BLAST+ (23 seconds on average) and 2.6 times faster than MMseqs2 with the same settings but no parallelism (8.4 seconds on average). Running times were correlated with the lengths of the queries, with a linear correlation of 0.57 for parallel MMseqs2 or 0.95 for BLAST.

BLAST+ and MMseqs2 use different alignment scores, so weak hits according to one scoring might not meet the E-value cutoff for the other, or they might be very far down in the sorted list of hits. To test the sensitivity of MMseqs2, we considered the top 3,188 BLAST+ hits for each query with $E \leq 10^{-5}$. (3,188 is half the number of genomes.) MMseqs2 with sensitive settings missed 1.4% or 2.2% of these hits (depending on the k-mer size), with a much lower miss rate for the top-ranked hits (Fig 1A). Fast parallel MMseqs2 missed 6.7% of hits, but most of these missed alignments were relatively weak: their median identity was 26.6%.

When examining gene prevalence, the focus is usually on proteins that are potential functional orthologs. (Focusing on orthologs also justifies considering only the top #genomes/2 hits: since relatively few proteins have orthologs in the majority of prokaryotes, hits further down in the list are unlikely to be orthologs.) We considered a hit to be a potential functional

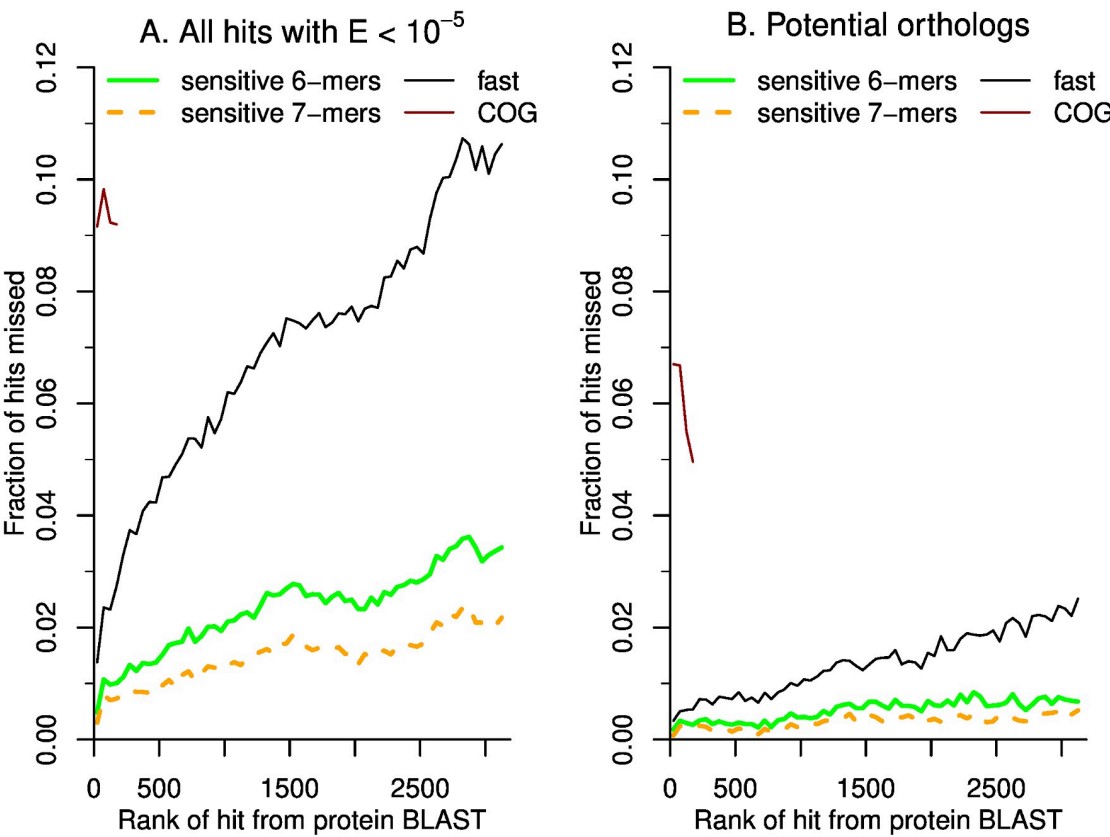

**Fig 1. Sensitivity of fast parallel MMseqs2 when searching against 6,377 representative genomes.** (A) The miss rate as a function of the hit's rank, for hits from BLAST+ with $E \leq 10^{-5}$. Each point is an average across 50 ranks and up to 1,000 queries. Besides fast parallel MMseqs2, we also show results for MMseqs2 with the most sensitive settings (using either 6-mers or 7-mers) and for COGs (as assigned using eggNOG-mapper). For COGs, we only considered the top 200 hits of each query. (B) The miss rate for potential orthologs (at least 30% identity and at least 50% coverage of the query).

ortholog if the alignment was at least 30% identity and covered at least 50% of the query. By this criterion, the average query had 1,847 potential orthologs (the median was 724). For potential orthologs with $E \leq 10^{-5}$ and rank $\leq$ #genomes/2, the miss rate of fast parallel MMseqs2 was just 1.2%. Again, the miss rate was much lower for the top-ranked hits (Fig 1B). Another threshold that is sometimes used for identifying potential orthologs is a bit score ratio of 0.3, that is, the bit score for the alignment is at least 30% of the maximum [21]. By this criterion, the average query had 893 potential orthologs (the median was 123). Among hits with $E \leq 10^{-5}$, rank $\leq$ #genomes/2, and score ratio $\geq$ 0.3, fast parallel MMseqs2 missed just 0.2% of hits.

Overall, by using parallel MMseqs2, *fast.genomics* can find homologs for a protein of interest across 6,377 genomes in a few seconds, with a sensitivity for potential orthologs of around 99%.

## Ortholog groups miss many potential orthologs despite being too broad

As discussed above, some fast comparative genomics tools rely on pre-computed ortholog groups from eggNOG. Given a protein of interest, eggNOG can quickly assign it to an ortholog group; if all of the proteins in the database have been assigned to ortholog groups, then this

assignment also yields a list of homologs. To test the sensitivity of this approach, we ran egg-NOG-mapper [22] on the top 200 homologs of each of our 1,000 test proteins.

We first considered the broadest ortholog groups returned by eggNOG-mapper, which are primarily from the COG database [23]. (COG is short for clusters of orthologous groups.) Of the 1,000 queries, 93 were not assigned to ortholog groups, and could not be handled by the ortholog approach. 40 of these lacked homologs besides themselves (as identified by BLAST + with E $\leq 10^{-5}$) but the other 53 did have at least one homolog in our top-level database, and in all of these cases, fast parallel MMseqs identified at least one homolog. Across the remaining 907 queries, the ortholog group approach missed 9.2% of the high-ranking homologs. If we restrict our attention to high-ranking potential orthologs (at least 30% identity and 50% coverage, and again, rank at most 200), ortholog groups still missed 6.0% of homologs. Some of these potential orthologs might have different domain content, which would lead them to be classified in a different COG (even if they are close homologs). But when we required the alignment to cover 90% of both the query and the subject, ortholog groups still missed 3.6% of homologs. For comparison, among these high-ranking high-coverage hits (that were assigned to ortholog groups by eggNOG-mapper), fast parallel MMseqs2 missed just 0.1% of homologs.

Another issue with pre-computed ortholog groups is that they are often too broad to be useful. For example, in the model gut bacterium *Bacteroides thetaiotaomicron* VPI-5482, 41% of genes are in top-level ortholog groups with 5 or more members. Lower-level ortholog groups are narrower and can avoid this problem, but they have a much higher rate of missed homologs. For example, the third-level ortholog group (usually at the phylum level) missed 26% of high-ranking potential orthologs (homologs with at least 30% identity, at least 90% coverage both ways, and rank at most 200). Overall, searching for homologs with fast parallel MMseqs2 is much more accurate than relying on ortholog groups.

## Speed and sensitivity of finding homologs in a large order

Because the largest order-level databases contain over ten million proteins each (Table 1), BLAST can be too slow for interactive analysis of these databases. *Fast.genomics* does not use MMseqs2 for the order-level databases because of MMseqs2's memory requirements (63 GB for the 22 million proteins in the main database). Instead, *fast.genomics* uses a strategy based on sequence clustering. The largest orders contain roughly 2–3 times more genomes than species, and over 10 times more genomes than genera. This suggests that these orders contain many highly-similar sequences. Indeed, when we used CD-HIT [24, 25] to cluster the proteins in each order at 70% identity and 90% coverage both ways, we reduced the number of sequences in the largest orders by 3.5- to 6.2-fold (Table 1).

Given the redundancy of the large orders, *fast.genomics* can find homologs quickly by first searching against the reduced database, using BLAST+, and then searching for additional homologs in each cluster, using LAST [26]. To test the speed and accuracy of these clustered searches, we selected 1,000 random proteins from the Rhizobiales, which is the order with the largest number of proteins, namely 12.5 million, which reduces to 2.4 million after clustering. We ran both clustered search and regular BLAST+ for 1,000 randomly-selected proteins from this order. All analyses used 12 threads and each query was run separately. Regular BLAST+ took an average of 11.7 seconds, while clustered BLAST took an average of 3.6 seconds (3.3 times faster).

To quantify the sensitivity of clustered BLAST, we considered the top #genomes/2 = 1,635 hits from regular BLAST+ with E $< 10^{-5}$. 1.8% of these homologs were missed by clustered BLAST. Among the top 200 homologs for each query, just 0.3% were missed. Among potential orthologs (30% identity and 50% coverage of the query), 1.0% were missed. Overall, clustered

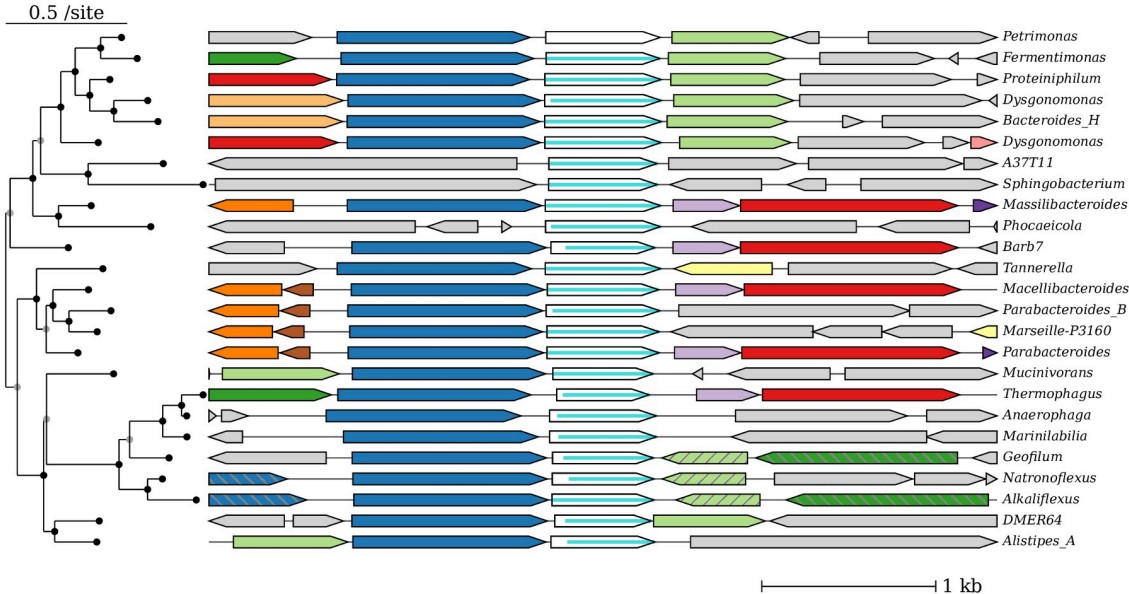

**Fig 2. The gene neighborhood view.** The query protein is ING2E5A_RS06865.

BLAST typically finds homologs within the largest order in a few seconds, with a sensitivity for potential orthologs of around 99%.

## Viewing gene neighborhoods

Once homologs have been identified for a protein of interest, *fast.genomics* can quickly show the gene neighborhoods around its top homologs (Fig 2). Hovering on each gene shows its annotation. A blue bar within each homolog shows the extent of homology. Hovering on the bar gives more information about the alignment, and clicking on it runs protein BLAST to show the pairwise alignment.

To show if gene neighbors are conserved, the genes are color-coded by homology. Specifically, *fast.genomics* runs LAST [26] on all of the protein sequences of genes that are visible. Then, genes that are similar (with at least 50% coverage) are assigned the same color. This usually takes less than a second.

By default, the genus or species is shown at the right side of each gene neighborhood, and hovering on the taxon shows the taxonomic lineage. Alternatively, the full taxonomic lineage can be shown above each gene neighborhood. This makes it easier to understand which taxa contain close homologs of the query, but fewer homologs can be seen at once.

By default, the homologs are shown in descending order of bit score, but the analyst can also request that a multiple sequence alignment and a phylogenetic tree be computed. *Fast. genomics* uses MUSCLE [27], FastTree 2 [28], and midpoint rooting; together, these usually take a second or less. Splits in the tree are reordered to show the closest homologs at the top (Fig 2). This feature of *fast.genomics* was inspired by the MicrobesOnline tree browser [29], but MicrobesOnline uses pre-computed trees and was last updated in 2011.

Because the order-level databases contain up to 10 representative genomes for each species, there are often multiple very-similar homologs from different representatives of a species. By default, the gene neighborhood viewer will collapse proteins that belong to the same species if they belong to the same CD-HIT cluster (at least 70% identity). This allows more distant

| Phylum | #Genomes | # with hits | #Hits | Max ratio |
|---|---|---|---|---|
| B Bacteroidota | 705 | 319 | 842 | 1.00 |
| B Planctomycetota | 158 | 84 | 127 | 0.29 |
| B Verrucomicrobiota | 110 | 51 | 96 | 0.27 |
| B Acidobacteriota | 111 | 25 | 35 | 0.28 |
| B Pseudomonadota | 1,922 | 28 | 31 | 0.35 |
| B Armatimonadota | 32 | 16 | 20 | 0.25 |
| B Gemmatimonadota | 18 | 12 | 17 | 0.26 |
| B Hydrogenedentota | 5 | 2 | 3 | 0.14 |
| B Myxococcota_A | 6 | 2 | 2 | 0.23 |
| B KSB1 | 2 | 1 | 1 | 0.25 |
| B Latescibacterota | 4 | 1 | 1 | 0.08 |
| B OLB16 | 3 | 1 | 1 | 0.12 |

**Fig 3. The taxonomic distribution view.** Here we show the prevalence of potential orthologs of ING2E5A_RS06865 across phyla. "B" at left is short for bacteria. "Max ratio" reports the highest bit score ratio (the alignment score for a homolog divided by the alignment score of the query protein to itself) among homologs from that taxonomic group.

homologs to be shown, and also makes it easy to see if close homologs are conserved within each species.

## Viewing the taxonomic distribution of a protein's homologs

*Fast.genomics* can show which taxa contain homologs of a protein, at whatever taxonomic level the analyst selects. By default, all homologs are considered, but *fast.genomics* can also consider only potential orthologs (at least 30% identity and 50% coverage), as in Fig 3, or "good" homologs (with a bit score at least 30% of the maximum score). Alternatively, the analyst can download a table of homologs that includes GTDB's classification of their genomes, bit scores, and e-values.

## Comparing the presence and absence of two proteins' homologs

Once two proteins are selected, and their homologs have been computed, *fast.genomics* can compare their distributions (co-occurrence analysis or phylogenetic profiling) in two ways. First, as shown in Fig 4, *fast.genomics* can plot the score ratio (the bit score divided by the maximum score) for the best hit in each genome. If the two proteins co-occur, then a genome with a high score ratio for one protein will tend to have a high score ratio for the other. *Fast.genomics* also highlights genomes that have the two proteins encoded close by (within 5 kb and on the same strand). This aspect of the presence/absence plot is conceptually similar to the information in the gene neighbor view, but the gene neighbor view does not scale to so many homologs. In this case, relatively weak homologs (score ratios below 0.2) are sometimes encoded close by. Fig 4 also shows a few high-scoring homologs that are in the same genome but not nearby.

*Fast.genomics* reports how often the homologs co-occur, both for all homologs and for "good" homologs with a bit score $\geq$ 30% of the maximum. *Fast.genomics* also selects a rank threshold that minimizes the probability of the observed co-occurrence, under a simple neutral model in which each gene has a fixed probability of being present in any genome. This

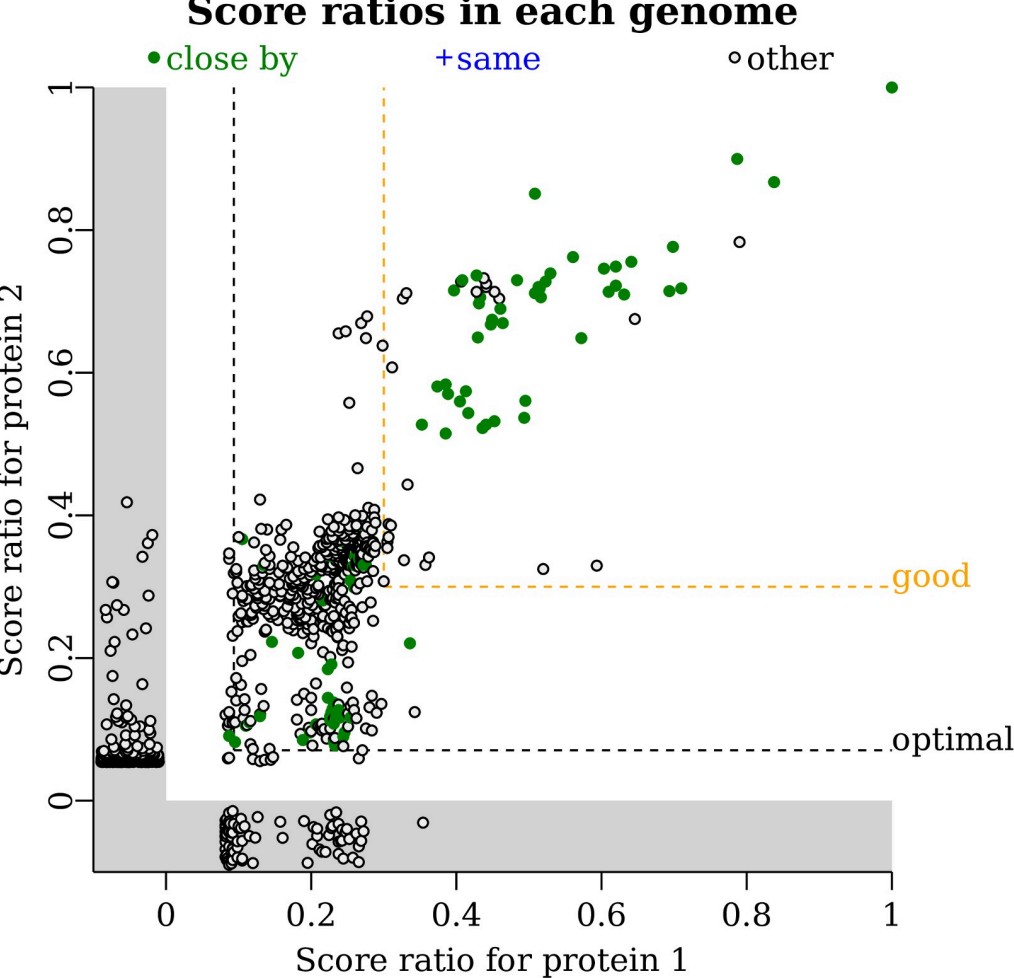

**Fig 4. The presence/absence plot.** Each point is a genome that contains a homolog of protein 1 and/or protein 2. Each axis shows the score ratio (the bit score divided by the maximum) for the highest-scoring homolog in that genome. Genomes are highlighted in green if the two homologs are encoded close by. If the same protein in a genome is the highest-scoring homolog for both queries (such as a fusion protein), then the genome would be highlighted in blue. Other genomes (with the homologs not encoded nearby) are shown in black. If a genome contains a homolog for one query but not the other, then the score is shown in the gray region below zero. Hovering on a point shows the taxonomic lineage of that genome. The two query proteins are ING2E5A_RS06865 and ING2E5A_RS06860.

threshold is shown as the "optimal" line (Fig 4). This feature was inspired by partial phylogenetic profiling [30], but to maintain a quick interactive response, *fast.genomics* optimizes one threshold (the same rank threshold for both proteins) instead of two thresholds. *Fast.genomics* uses Fisher's exact test with a Bonferonni correction for the number of possible thresholds considered.

Although the presence/absence plot summarizes information from many genomes, it does not give a quick indication of which taxa contain the genes. *Fast.genomics* can also show a table of the taxa that contain both genes, or of the taxa that contain the two genes in proximity.

## Links to other sequence analysis tools

Although *fast.genomics* does not include pre-computed sequence features such as protein family membership, the protein page includes links to a variety of sequence analysis tools. These

include two fast ways to place proteins into families: searching against the Conserved Domain Database [31], or finding the closest homolog in UniProt using SANSparallel [32], which then links to the InterPro results for that homolog [33]. In our experience, most of the proteins in the *fast.genomics* database have a homolog with 99%-100% identity in UniProt.

## Limitations of fast.genomics

The main limitation of *fast.genomics* is probably the splitting of the database by order. This might make *fast.genomics* less effective for selfish genetic elements that can move between distantly related bacteria. Still, the main (genus-level) database can quickly indicate which orders frequently contain the protein of interest.

To ensure that presence/absence analyses give reliable results, *fast.genomics* includes only high-quality genomes; but this means that much of the diversity visible in metagenomes is missing from *fast.genomics*. For instance, of the 20,739 genera in GTDB, only about a third (6,377) are included in *fast.genomics*. Including low-quality metagenome-assembled genomes might allow analysts to identify more conserved gene neighborhoods. Still, the *fast.genomics* main database includes 1,418 genomes from uncultivated genera, of which 1,368 are metagenome-assembled genomes. And most prokaryotic proteins do have homologs in the main database: for instance, for random proteins from its database, *fast.genomics* finds at least one potential ortholog (at least 30% identity and 50% coverage) for 94% of queries.

*Fast.genomics* does not support iterative searches to find distant homologs (homologs that are too distant to be found using pairwise methods such as BLAST). In this case, we recommend using the HMMer web server [34] for iterated searches against reference proteomes, or using a tool such as AnnoTree [15] for predefined families.

## Comparisons to other comparative genomics websites

As far as we know, the other fast websites for comparative genomics of bacteria and archaea all rely on pre-computed orthology groups, such as eggNOG. Above, we assessed the accuracy of ortholog groups for finding homologs. Here, we give an example of how tools that rely on ortholog groups can fail. Consider the TonB-dependent transporter BT2172. We previously proposed, based partly on comparative genomics evidence, that this transporter functions together with another protein, BT2173, belonging to the uncharacterized family DUF4249 [35]. *Fast.genomics* shows that these proteins form a conserved operon, and that this subfamily of TonB-dependent transporters is almost always encoded next to a DUF4249 protein, across dozens of genera.

When we ran GeCoViz [14] with the sequence of BT2172 and selected the top-scoring eggNOG (ENOG501R5B9, E = 3.94e-91), GeCoViz failed to identify any highly conserved gene neighbors. Specifically, no neighbors met the conservation threshold of 0.5. When we dropped the threshold to 0.1, then a variety of conserved gene neighbors were reported, but not DUF4249. This is because the top-scoring eggNOG mixes BT2172 and its close homologs together with several other larger subfamilies of TonB-dependent receptors.

We also analyzed BT2172 using the STRING database of functional associations between proteins, which includes associations based on conserved proximity or co-occurrence. These associations are pre-computed using a combination of BLAST searches and orthology groups [36, 37]. (As of October 2023, STRING 12.0 includes 2,424 core genomes and 12,535 total genomes, and all proteins from core genomes are compared to all proteins from all genomes.) Given BT2172 (BT_2172) as a query, STRING correctly identified BT2173 (BT_2173) as a conserved gene neighbor. However, STRING also reported that the co-occurrence of the two proteins across genomes was "none / insignificant." STRING's co-occurrence viewer suggested

that BT2172 is present in diverse bacteria, while BT2173 is present only in Bacteroidetes. In contrast, *fast.genomics* reports that the closest homologs of the two proteins co-occur ($P = 10^{-23.6}$). Our interpretation is that the distant homologs of BT2172 from other phyla have a different function. STRING's co-occurrence viewer does show that the homologs of BT2172 from other phyla are quite distant, but by a subtle color coding, and this is apparently not taken into account in the automated analysis. In reality, disrupting either BT2172 or BT2173 has similar consequences [35], which confirms that they function together and suggests that the distant homologs of BT2172 from genomes that lack homologs of BT2173 have a different function. But in the absence of the genetic data, the report from STRING that BT2172 has a different distribution than BT2173 would suggest that BT2172 can function on its own, which is misleading.

Conversely, BLAST-based tools should give accurate results, but they are far slower than *fast.genomics*. Based on a recent review [38], we identified three websites that are still maintained and that report gene neighborhoods for the most similar proteins, as identified using BLAST. These were WebFlaGs, the Enzyme Function Initiative's Genome Neighborhood Tool (EFI-GNT retrieve neighborhood diagrams), and IMG/M (top homologs combined with gene cart neighborhoods) [39–41]. When we ran these tools using a 290 amino acid protein (BT2157) as the query, the quickest response was from webFLaGs, in 11 minutes. None of these tools show the gene neighborhoods together with information about the similarity of each homolog or the extent of homology (in case of changes to domain structure). In the IMG/M results, the first 60 homologs are over 90% identical, and all but one of these is from the genus *Bacteroides*. Information from such close homologs is probably not functionally informative, because gene neighborhoods beyond operons (that is, the proximity of genes that probably lack a functional relationship) are often conserved in closely-related bacteria [42]. WebFlaGs and EFI-GNT have options to search against reduced databases (such as UniRef90, which is clustered at 90% identity), which avoids this problem. Similarly, *fast.genomics* avoids this issue by having just one representative per genus in the main database, or for the sub-databases, by clustering together similar proteins from the same species.

As far as we know, none of these websites can use a tree derived from protein sequences to organize the results. (This capability is described in the WebFlaGs manuscript, but as of October 2023, it is not available.) By grouping together similar sequences, *fast.genomics* can highlight subfamilies with consistent gene neighborhoods. In contrast, GeCoViz and STRING use the species tree to organize their displays. Because of rampant horizontal gene transfer, the species tree is generally not a reasonable guide to the gene tree. Furthermore, neither GeCoViz nor STRING's gene neighborhood viewer shows which homologs are closely related to the query.

## Conclusions

*Fast.genomics* supports interactive browsing of protein families across the diversity of bacteria and archaea. To identify homologs for a protein of interest in a few seconds, *fast.genomics* splits the database of genomes into a main database, with one representative per genus, and a database for each taxonomic order. Furthermore, *fast.genomics* uses accelerated searches which are almost as sensitive as protein BLAST, but are several times faster. These accelerated searches are much more accurate than pre-computed ortholog groups. Once homologs have been identified, *fast.genomics* can rapidly show their gene neighborhoods and their taxonomic distribution.

Given ongoing improvements in long-read sequencing and metagenomics, we expect the number of genera with high-quality genomes to increase substantially over the next few years.

For example, if all of the genera in GTDB had at least one representative with a high-quality genome, then *fast.genomics*' main database would need to expand three-fold. In the future, we hope to maintain quick access to the expanding diversity of high-quality prokaryotic genomes by using faster CPUs and solid-state storage and by making improvements to the parallelism of MMseqs2.

## Materials and methods

### Data sources

Taxonomic assignments and metadata about assemblies, including CheckM scores and MIMAG quality assessments, were downloaded from the Genome Taxonomy Database (release 08-RS214). Assemblies, including gene annotations, were downloaded from RefSeq or Genbank. Although we ran GUNC on some assemblies ourselves, for the most part, we relied on previous classifications of assemblies as being non-chimeric, either from the GUNC website (https://grp-bork.embl-community.io/gunc/datasets.html) or from proGenomes3 [43].

### Which genomes to include

As mentioned above, we required that each genome have CheckM completeness of at least 90% and contamination of at most 5%; we required that the genome not be classified as chimeric by GUNC; and if the genome was not from an isolate, we required that it meet the MIMAG criteria for high quality. Assemblies without protein annotations in Genbank or Refseq were not considered. Also, we excluded a few genomes that had an unreasonably high proportion of pseudogenes; specifically, we required that at least half of the genes be protein coding.

We selected one genome per genus for the main database, and up to 10 genomes per species for the order-level databases. When selecting genomes, we preferred assemblies that were in RefSeq; that were the type species of a genus (according to GTDB); that were selected by GTDB as the representative of a species; that had lower 2 * contamination—completeness (using CheckM scores); that were in proGenomes3's list of highly important strains; or that had a longer largest scaffold.

### Fast parallel MMseqs2

MMseqs2 provides an "easy-search" workflow that emulates the output of protein BLAST. This workflow includes four steps: createdb, prefilter, align, and convertalis. Most of the time is taken up by the prefilter step, which finds potential homologs with a promising ungapped alignment, and the align step, which searches around each of those ungapped alignments for the optimal gapped alignment. We wrote a perl script (mmseqsParallel.pl) which runs the same steps as easy-search, but splits the results of the prefilter into 10 parts. It then runs the align step on those 10 parts in parallel, combines the parts, keeps the top #genomes entries, and runs the final step. mmseqsParallel.pl gives identical results as MMseqs2's easy-search except for the the truncation of the result list to #genomes entries and the ordering of hits with the same e-value and bit score.

For the typical query, with no parallelization, the prefilter and align steps each take about the same amount of time, which explains why the median time for parallel MMseqs2 is about half that for regular MMseqs2 (6.1 seconds versus 3.0 seconds). However, for longer queries that have many homologs, the align step can take much longer, so that parallel MMseqs2 gives a somewhat larger reduction in average time (2.6x instead of 2.0x).

As an alternative to running the alignment step in parallel, we also tried splitting the database into 10 pieces, and running MMseqs2 in parallel on each piece. This was not as fast as

parallelizing the alignment step only because the prefilter step is I/O intensive, and splitting the database into 10 pieces would increase the number of random memory accesses against the index of k-mers by 10-fold.

Conceptually, the prefilter stage combines two separate computations: looking up k-mers to find candidate homologs (and which diagonal the homology is on), and finding the best ungapped alignment for these candidates. Looking up k-mers is I/O intensive and probably wouldn't benefit from using multiple CPUs, but computing ungapped alignments might benefit from multi-threading. This is an opportunity to further speed up parallel MMseqs2.

## Settings for MMseqs2

We chose to use a k-mer size of 6 instead of 7, as this leads to only a slight decrease in the maximum sensitivity (Fig 1) and it reduces the time in the prefilter step, which cannot be parallelized. For a k-mer size of 6 and 6,377 genomes, the MMseqs2 database is 63 GB (including the k-mer index) and takes about 5 minutes to build.

To run MMseqs2 with sensitive settings, we used the maximum value of the sensitivity parameter (7.5) and asked MMseqs2 to consider up to 100 * #genomes alignments (the max-seqs parameter). Fast parallel MMseqs2 considers up to 8 * #genomes alignments and returns a maximum of #genomes alignments. Fast parallel MMseqs2 varies the sensitivity parameter depending on the query length because in initial testing, the miss rate was much higher for shorter sequences. The sensitivity levels are: 7.5 (the maximum) for queries of up to 150 amino acids; or 7.0 if the query is 250 amino acids or less; or 6.25 if 350 amino acids or less; or 6.0 if 650 amino acids or less; or 5.7 (the default) for queries of over 650 amino acids. With these settings, the miss rate for potential orthologs is still the highest for the shortest queries (2.0% for queries of 150 amino acids or less), but these are already being run at maximum sensitivity.

Because MMseqs2 with sensitive settings returns up to 100 * #genomes alignments, while fast parallel MMseqs2 returns only #genomes alignments, the difference in their sensitivity (shown in Fig 1) is inflated, but only by a small amount. If we consider only the top #genomes hits from MMseqs2 with sensitive settings and a k-mer size of 6, then its miss rate for potential orthologs increases slightly from 0.6% to 0.7%. This difference reflects the different scoring used by BLAST+ and MMseqs2, which leads to a different ordering of the hits.

## Clustered search

Given a query, clustered search runs BLAST+ against a database of cluster representatives. Then it expands the hits by adding additional members of the clusters, and it uses LAST [26] to compare the query to all of the candidate homologs. To limit the time for the second step, the number of potential homologs is limited to the number of genomes in the order-level database or 200, whichever is greater. Note that LAST and BLAST+ report different bit scores.

The clusters were identified by running CD-HIT [24, 25] separately for each order-level database. We ran CD-HIT with 5-mers, required 70% identity and 90% alignment coverage, and used 50 threads (-n 5 -c 0.7 -aS 0.9 -aL 0.9 -T 50). CD-HIT is the slowest part of building the *fast.genomics* order-level databases, which took a total of 44 hours.

## E-values and compositional bias

All homology searches in *fast.genomics* use a e-value cutoff of $10^{-3}$. For searches against the main database, e-values and alignment scores (bit scores) are from MMseqs2, which adjusts for local compositional bias [17]. The e-values from MMseqs2 should be unbiased: in a large-scale validation test that was conducted by its authors, and that included proteins with

disordered and low-complexity regions, about 1 per 1,000 queries had any false positives with $E < 10^{-3}$ [17]).

For searches against the order-level databases, the alignment scores and e-values are from LAST [26], which does not correct for compositional bias. We do not expect the use of LAST to add false positives because proteins are only considered by LAST if they are very similar to homologs that were identified using BLAST+, which does correct for compositional bias [44]. But, the authors of MMseqs2 found that e-values from BLAST+ are overly optimistic [17], so we wondered if there was a high rate of false positives in the clustered search results. For the 1,000 random query proteins from Rhizobiales, we considered the weak hits from clustered search ($10^{-5} < E < 10^{-3}$). We considered only the top 100 hits for each query, because by default, the gene neighborhood view shows the top 50 species clusters, and this database has roughly twice as many genomes as species. 85% of the high-ranking weak hits were found by MMseqs2 as well (MMseqs2 e-value < 0.05) and are unlikely to be false positives. We checked five of the exceptions and all five pairs are probably true homologs: they matched the same family (hidden markov model) in InterPro [33] or they had similar predicted structures [45].

Regardless of the risk of false positives, distant homologs (under 30% identity) are likely to have different functions, and should probably be ignored unless they have conserved gene neighborhoods.

## Features of the gene neighborhood view

To identify sequence similarity between the genes that are shown, *fast.genomics* runs LAST [26] on their protein sequences, using 12 threads and default settings. Only alignments that cover at least 50% of both sequences are considered. A greedy clustering algorithm is used to group similar genes together, and each group is given the same color. Two hatch patterns are used to expand the effective number of colors from 11 colors to 33. The query and its homologs are shown in white, which is effectively another color.

To infer a phylogenetic tree, MUSCLE 3 is run with fast settings (-maxiters 2 -maxmb 1000) and only the part of each sequence that is similar to the query is included in the alignment. Then, FastTree 2 is run with default settings, followed by rooting the tree at the midpoint (to minimize the maximum distance from the root to any sequence).

*Fast.genomics* can show up to 200 homologs. (Fig 1 shows only 25 homologs to save space.) But some genes have thousands of potential orthologs. To make it easier to see the range of gene neighbors for a protein family, *fast.genomics* can select homologs (or "good" homologs with bit score $\geq$ 30% of the maximum score) at random, instead of showing the top homologs.

Genes in the same operon often have a functional relationship, and genes are much more likely to be in the same operon if they are very close together [46, 47]. So it can be important to know just how close the genes are together—but this is difficult to see in a gene-level view. In *fast.genomics*, hovering in between two genes or near the edge of a gene shows the spacing in nucleotides.

## Implementation of the web site

The web site is implemented in perl 5 using the common gateway interface (CGI). Graphics are rendered using scalable vector graphics (SVG). To avoid redoing computations such as identifying homologs for a protein sequence, color coding the proteins in the gene neighborhood view, or inferring a phylogenetic tree, the inputs are hashed using MD5, which produces a 128-bit hash, and the results are stored on disk based on that hash.

Tables of genomes, genes, proteins, and taxa are stored in a SQLite3 database. There is a separate SQLite3 database file for the main database and for each order, so that most of the code is the same regardless of whether it is operating on the main database or an order-level database.

## Hardware and memory requirements

The *fast.genomics* server has 1 TB of memory and 64 AMD Opteron 6376 CPUs running at 1.4 GHz. The server is used for other memory-intensive computations as well, but it usually has little load.

For the top-level database, which has 21.8 million proteins and 7.1 billion amino acids, the MMseqs2 database requires 63 GB. For good performance, this needs to be kept in memory. A SQLite3 database of genes takes another 5.2 GB and this should also be kept in memory. For example, to describe the taxonomic prevalence of a protein's homologs, *fast.genomics* needs to look up the gene(s) and genome identifier(s) for every homolog's protein identifier. For clustered search within each order, the clustering information (18.5 GB) needs to be kept in memory, as looking up the additional members of thousands of clusters could otherwise take several seconds. Overall, the server needs at least 87 GB of memory. We run a script, once per hour, to read these tables or files to ensure that they are kept in memory. In the future, we hope to switch to solid state storage, and this then will be less important.

## Author Contributions

**Conceptualization:** Morgan N. Price, Adam P. Arkin.

**Data curation:** Morgan N. Price.

**Formal analysis:** Morgan N. Price.

**Funding acquisition:** Adam P. Arkin.

**Investigation:** Morgan N. Price.

**Project administration:** Adam P. Arkin.

**Resources:** Adam P. Arkin.

**Software:** Morgan N. Price.

**Supervision:** Adam P. Arkin.

**Writing – original draft:** Morgan N. Price.

**Writing – review & editing:** Morgan N. Price, Adam P. Arkin.

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
