## [Decision Letter · Decision Letter 0]

8 Feb 2024

PONE-D-23-38656A fast comparative genome browser for diverse bacteria and archaeaPLOS ONE

Dear Dr. Price,

Thank you for submitting your manuscript to PLOS ONE. After careful consideration, we feel that it has merit but does not fully meet PLOS ONE’s publication criteria as it currently stands. Therefore, we invite you to submit a revised version of the manuscript that addresses the points raised during the review process.

We look forward to receiving your revised manuscript.

Kind regards,

Vasilis J Promponas

Academic Editor

PLOS ONE

Journal Requirements:

"This material by ENIGMA- Ecosystems and Networks Integrated with Genes and

Molecular Assemblies (http://enigma.lbl.gov), a Science Focus Area Program at

Lawrence Berkeley National Laboratory is based upon work supported by the

U.S. Department of Energy, Office of Science, Office of Biological &

Environmental Research under contract number DE-AC02-05CH11231"

"This material by ENIGMA- Ecosystems and Networks Integrated with Genes and Molecular

Assemblies (http://enigma.lbl.gov), a Science Focus Area Program at Lawrence Berkeley

National Laboratory is based upon work supported by the U.S. Department of Energy, Office of

Science, Office of Biological & Environmental Research under contract number

DE-AC02-05CH1123."

"This material by ENIGMA- Ecosystems and Networks Integrated with Genes and

Molecular Assemblies (http://enigma.lbl.gov), a Science Focus Area Program at

Lawrence Berkeley National Laboratory is based upon work supported by the

U.S. Department of Energy, Office of Science, Office of Biological &

Environmental Research under contract number DE-AC02-05CH11231"

6. Please update your submission to use the PLOS LaTeX template. The template and more information on our requirements for LaTeX submissions can be found at http://journals.plos.org/plosone/s/latex.

**Additional Editor Comments:**

Specifically, the reviewers who assessed your manuscript and the Fast.genomics software/server (whose detailed comments can be found below) both find value in your work: Fast.genomics is a welcome addition to the computational tools available for computational comparative genomics, functions as described in the paper and the manuscript is clearly written.  

Reviewer #1 poses several interesting topics for discussion, which I believe are easy to address in a revised version of this work. Of particular importance is the comment on the suitability of different e-value thresholds for establishing homology relations; it can be emphasized that by providing the Fast.genomics data and code in GitHub, interested researchers can experiment and fine tune the tools for their particular needs.

Reviewer #2 initially highlights the lack of comparison to online compute environments providing similar functionality to Fast.genomics. Even though it might be of interest to see detailed benchmarks to systems like IMG or MGnify, I believe it will be adequate to only provide high level comparisons (e.g., comparison of features available) since such systems are installed in compute infrastructures which are not easy to replicate/have access to and their performance when run over the web can vary based on the specific workload of their servers. Regarding the second comment, although this is definitely worth discussing within the manuscript, I understand that providing a detailed comparison to the work presented in the recently published paper by Pavlopoulos and colleagues is a whole project on its own. However, adding some practical tips (either in the manuscript or in the GitHub repository) on how interested readers could exploit these newly characterized protein families for enhancing Fast.genomics functionalities would be welcome.

On a personal note, it has long been reported that local compositional extremes (e.g., regions with low complexity/compositional bias/tandem repeats) in protein sequences can be a major confounding factor when searching for homologous proteins. However, by carefully inspecting the manuscript//webserver and quickly going through the GitHub repository, I could see no information on how your approach deals with such cases. I suggest that the manuscript text is amended to accurately reflect on how local compositional biases are handled and an appropriate rationale for this choice. Even though such features are not as prominent in prokaryotes, a simple UniProt query (https://www.uniprot.org/uniprotkb?query=%28taxonomy_id%3A83333%29+COMPBIAS; accessed Feb 05, 2024) yields more than 200 proteins annotated with compositional bias in the Escherichia coli (strain K12) proteome. It would be illuminating to examine how Fast.genome handles such proteins, especially with respect to identification of spurious (i.e. false positive) homologs.

Reviewers' comments:

Reviewer's Responses to Questions

**Comments to the Author**

1. Is the manuscript technically sound, and do the data support the conclusions?

Reviewer #1: Yes

Reviewer #2: Yes

2. Has the statistical analysis been performed appropriately and rigorously? 

Reviewer #1: Yes

Reviewer #2: Yes

3. Have the authors made all data underlying the findings in their manuscript fully available?

Reviewer #1: Yes

Reviewer #2: Yes

4. Is the manuscript presented in an intelligible fashion and written in standard English?

Reviewer #1: Yes

Reviewer #2: Yes

5. Review Comments to the Author

Reviewer #1: The authors have developed a web-based tool, called fast.genomics, that supports fast browsing of prokaryotic genomes.

They have a main database that contains one representative from each of the 6,377 genera that have high quality genomes. They also have a second database that contains 10 representatives of each species.

Next, they identify homologs of proteins of interest. Afterwards, fast genomics shows their distribution in the various taxa, as well as their neighboring genes. Furthermore, the software can compare the distribution of homologs for two different proteins. The main idea behind this is to predict function, via phylogenetic profiling. Also, such an approach may identify operons that may be horizontally transferred in diverse taxa.

This is a very useful prokaryotic comparative genomics tool that focuses on rapid analysis and allows for quick conclusions. The web-site works fine and the underlined software that the pipeline relies on for sequence analysis and phylogeny are all standard tools well accepted by the community.

Reassuringly, the authors first apply quality filters to the genomes that they integrate.

Also, the manuscript is well written and easy to follow.

I only have some minor corrections/comments for future updates of the software.

One computational limitation is rapid identification of homologs in such vast datasets of thousands of genera and species. Thus, the authors use MMseqs2, which is a fast alternative to Blast. How does MMSeqs2 compare to DIAMOND (Buchfink et al., 2015, PMID: 25402007)? From DIAMOND’s web site “DIAMOND is a high-throughput program for aligning DNA reads or protein sequences against a protein reference database such as NR, at up to 20,000 times the speed of BLAST, with high sensitivity.”

The authors apply a homology cutoff of 1e-3. From my experience, I see better results with 1e-5. 1e-3 tends to get a lot of false positives, but this is only a comment of mine. Since different researchers have their own criteria for homology, depending on what they are after, it would be good to have a local version where the user may define their sets of genomes and their homology criteria, such as e-value and % identity with certain coverage.

Another issue is the criteria used for orthology. This is much more difficult than identifying homologs. The authors justify their own criteria for a fast search, which is the main idea behind this work. Otherwise, better and more accurate approaches would not allow for such large-scale and rapid analyses. Thus, the orthology criteria constitute a reasonable tradeoff between precision and recall.

Concerning which genomes to include: I understand that the authors use the species name as provided by genebank. However, very frequently, genomes are misannotated in terms of species names (Nikolaidis et al., 2022 and Nikolaidis et al., 2023 - PMID: 36144322 and PMID: 37266990). One approach, maybe for future updates of the web-tool is to use FASTANI (Jain et al., 2018, PMID: 30504855), to include the proper genomes to each species. However, it would be good to discuss this issue of species misannotations in Genbank (which is caused by the researchers that submit their genomes).

Reviewer #2: The authors propose a method for identifying homologs of protein(s) of interest across many genomes. Focussing on speed-up, they argue that their approach is faster than standard approaches such as protein BLAST or similar tool to find homologs, that, given the size of current genome databases, typically require over 10 minutes. They argue against alternative approaches that pre-compute groups of homologous proteins, using resources such as PFam (for examples, fast tools that rely on pre-computed homology groups such as GeCoViz, AnnoTree and PhyloCorrelate), as the authors believe they may give misleading results, especially if the protein of interest is a poor match to the pre-computed families.

The authors have built an interactive web service for browsing of protein families (fast.genomics - https://fast.genomics.lbl.gov/). Moreover, they have taken multiple shortcuts to speed up the process of obtaining comparative analysis for protein(s) of interest. These include:

1) Selection of one high-quality representative genome for each genus, so the main database of fast.genomics contains fewer genomes (6,377 genomes) to be used as a search space. They also allow for the expansion to up to 10 representatives of each species if the initial database does not provide adequate search results.

2) The use of a parallel version of MMseqs2 that allows for splitting of candidates from the prefilter step of MMseqs2 into 10 lists and analyzes them in parallel.

3) Use of CD-HIT to cluster the proteins in the largest order-level databases to speed-up

To test the speed and sensitivity of their approach, the authors selected 1,000 proteins at random from the main database and searched for homologs of these proteins in the main database using protein BLAST from NCBI’s BLAST+ package, fast parallel MMseqs2, or MMseqs2 with the most sensitive settings.

Their result showed that Fast parallel MMseqs2 took an average of 3.3 seconds per query, which was 7 times faster than BLAST+ (23 seconds on average) and 2.6 times faster than MMseqs2 with the same settings but no parallelisation (8.4 seconds on average).

The also asses the sensitivity of their parallel MMseq2 approach by considering the top 3,188 BLAST+ hits for each query with E ≤ 10-5. MMseqs2 with sensitive settings missed 1.4% or 2.2% of these hits (depending on the k-mer size), with a much lower miss rate for the top-ranked hits. Fast parallel MMseqs2 missed 6.7% of the hits.

The authors also compared to some fast comparative genomics tools that rely on pre-computed ortholog groups from eggNOG. They considered the broadest ortholog groups returned by eggNOG-mapper, for the 1,000 benchmark queries and found that 93 were not assigned to ortholog groups, and could not be handled by the ortholog approach. 40 of these lacked homologs besides themselves (as identified by BLAST+ with E ≤ 10-5) but the other 53 did have at least one homolog in their top-level database, and in all of these cases, fast parallel MMseqs identified at least one homolog. Across the remaining 907 queries, the ortholog group approach missed 9.2% of the high-ranking homologs. Among these high-ranking high-coverage hits (that were assigned to ortholog groups by eggNOG-mapper), fast parallel MMseqs2 missed just 0.1% of homologs.

The authors also report a reduction of the number of sequences in the largest orders by 3.5- to 6.2-fold using the CDHIT approach and also fast.genomics can find homologs quickly by first searching against the reduced database and at high sensitivity, again by comparing using a 1000 random protein benchmark dataset.

Fast.genomics further allows for extra functionalities including: 1) viewing gene neighborhoods, 2) viewing the taxonomic distribution of a protein’s homologs, 3) comparison of the distributions of two selected proteins, and their homologs (co-occurrence analysis or phylogenetic profiling) and 4) providing links to a variety of sequence analysis tools from the fast.genomics protein page.

The authors provide a specific example to highlight discrepancies that can arise from other fast websites for comparative genomics of bacteria and archaea that rely on pre-computed orthology groups, such as eggNOG. By considering the TonB-dependent transporter BT2172 and using GeCoViz with the sequence of BT2172 selected, the authors show that this tool failed to identify any highly conserved gene neighbours.

The authors also compare to BLAST-based tools that should give accurate results, but they are far slower than fast.genomics. The specifically compare with WebFlaGs, the Enzyme Function Initiative’s Genome Neighborhood Tool (EFI-GNT retrieve neighborhood diagrams), and IMG/M (top homologs

combined with gene cart neighborhoods). Again, they use the 290 amino acid protein (BT2157) as the query, the report the quickest response was from webFLaGs, at 11 minutes. The further showcase the functionalities of fast.geneomics by comparing against the 3 BLAST-based tools and show that none of these tools show the gene neighborhoods together with information about the similarity of each homolog or the extent of homology.

Overall, this is a nice approach that provides some interesting results with respect to speed-up of homology searching for metagenomic proteins as well as a user interface with some useful functionalities. However, I have the following major concerns:

1) I feel that the authors have not performed a complete and adequate comparison with other competing tools. Especially with some tools that provide an integrated environment for analysis, management, storage, and sharing of metagenomic projects, like IMG/M or MGnify (supported by the EBI). Although, they perform some form of assessment using a specific example, namely the BT2172 sequence, I would have liked to see a more thorough comparison with tools such as IMG/M. These tools are specifically built for running large jobs and are hosted in servers with high calibre specifications suited for fast running and high sensitivity of results.

2) Recently Pavlopoulos et al. Nature, 2023 (Unraveling the functional dark matter through global metagenomics) published an approach that examines functional diversity beyond what was currently possible through the lens of reference genomes. Their computational approach generates reference-free protein families from the sequence space in metagenomes. They analysed over 26,000 metagenomes and identified >1 billion protein sequences with no prior similarity to any sequences from >100,000 reference genomes or the Pfam database. It would be nice for the authors to make a comparison with this approach and provide at least some measure of sensitivity with the results reported by this publication. Moreover, Pavlopoulos et al. provided a tremendous amount of novel protein families which have not been considered by the authors in their approach.

6. PLOS authors have the option to publish the peer review history of their article (what does this mean?). If published, this will include your full peer review and any attached files.

Reviewer #1: No

Reviewer #2: No

---

## [Author Response · Author response to Decision Letter 0]

27 Feb 2024

Responses to comments from reviewer #1

Reviewer 1: "One computational limitation is rapid identification of homologs in such vast datasets of thousands of genera and species. Thus, the authors use MMseqs2, which is a fast alternative to Blast. How does MMSeqs2 compare to DIAMOND (Buchfink et al., 2015, PMID: 25402007)? From DIAMOND’s web site "DIAMOND is a high-throughput program for aligning DNA reads or protein sequences against a protein reference database such as NR, at up to 20,000 times the speed of BLAST, with high sensitivity."

DIAMOND is optimized for searches with many queries. For searches with a single query, as would occur during interactive use, our experience is that DIAMOND takes around the same time as protein BLAST.

Reviewer 1: "The authors apply a homology cutoff of 1e-3. From my experience, I see better results with 1e-5. 1e-3 tends to get a lot of false positives, but this is only a comment of mine."

The editor was also curious about this question, and the related issue of how fast.genomics handles compositional bias. To address it, we added a section to the Materials and Methods, titled "E-values and compositional bias". Briefly, we do not believe that there are many false positives in the hits with e-values near the cutoff.

We'd also like to note that for most queries, changing the e-value cutoff from 1e-3 to 1e-5 would not affect the gene neighborhood view with default settings. For the main database, the gene neighborhood view includes the top 50 hits by default. In the test set of 1,000 random prokaryotic proteins, only 8% have any hits from MMseqs2 with rank <= 50 and E > 1e-5 (and E < 1e-3). For the test set of 1,000 proteins from Rhizobiales, only 6% have hits from clustered search with rank <= 100 and E > 1e-5. (The default view for order databases shows species clusters of genes, and there's ~2x more genomes than species in this order-level database, so it is more appropriate to consider the top 2*50 = 100 homologs.)

More broadly, the focus of fast.genomics is on homologs which are likely to have the same function, namely closer homologs or homologs that have the same gene context (and are unlikely to be false positives).

Reviewer 1: "Since different researchers have their own criteria for homology, depending on what they are after, it would be good to have a local version where the user may define their sets of genomes and their homology criteria, such as e-value and % identity with certain coverage."

Fast.genomics allows the user to download a table of homologs. We revised the Results to mention that this table includes e-values. Similarly, the gene presence/absence tool allows the user to download a table of the top hit for both queries in each genome; this table now includes the e-values. In either case, the user could easily filter out weak hits from these tables, or choose a subset of genomes of interest.

If a user wishes to build a version of fast.genomics with additional genomes, the source code for fast.genomics is available, including the scripts for building the database. We revised the code availability statement to make it clear that the scripts for building the databases are included.

Reviewer 1: "Concerning which genomes to include: I understand that the authors use the species name as provided by genebank. However, very frequently, genomes are misannotated in terms of species names (Nikolaidis et al., 2022 and Nikolaidis et al., 2023 - PMID: 36144322 and PMID: 37266990). One approach, maybe for future updates of the web-tool is to use FASTANI…"

Actually, all taxonomic assignments in fast.genomics, including the species names, are taken from GTDB, which uses ANI comparisons to define species. We revised the section of the Results on "The fast.genomics databases" to explain that the species definitions in fast.genomics are from GTDB.

Responses to comments from reviewer #2

Reviewer 2: "1) I feel that the authors have not performed a complete and adequate comparison with other competing tools. Especially with some tools that provide an integrated environment for analysis, management, storage, and sharing of metagenomic projects, like IMG/M or MGnify (supported by the EBI). Although, they perform some form of assessment using a specific example, namely the BT2172 sequence, I would have liked to see a more thorough comparison with tools such as IMG/M. These tools are specifically built for running large jobs and are hosted in servers with high calibre specifications suited for fast running and high sensitivity of results."

In regards to "These tools are ... suited for fast running and high sensitivity of results", this is not our experience. For instance, the IMG/M web site states that "Real time BLAST request on average takes about 2 mins. to 15 mins. to complete." The homologs feature of IMG/M is just not designed to be fast. And, as discussed in our manuscript, we feel that the IMG web site does not organize the results as well as fast.genomics does. Similarly, our impression is that MGnify is not suitable for fast searches. As of February 13, the sequence search page says: "We recognize that our service has faced challenges in providing the latest version of the MGnify protein database, and we sincerely apologize for any inconvenience caused. The recent rapid growth of the protein database to over 3 billions has present technical challenges in scaling the search infrastructure which we are currently addressing."

We did not compare fast.genomics to MGnify because MGnify does not provide analogous functionality. In particular, as far as we know, MGnify does not provide any way to compare the gene neighborhoods of the homologous proteins or to compare the presence/absence of two proteins across taxa.

More broadly, the reviewer was concerned that we did not compare fast.genomics to tools that support metagenomics projects -- but supporting metagenomics might not be compatible with the goals of fast.genomics. In particular, fast.genomics includes only high-quality genomes (whether from isolates or assembled from metagenomes) to ensure that analyses of the presence or absence of a gene family, across genomes will give reliable results. It's not clear how to compare gene presence/absence across taxa from fragmented metagenomic assemblies. We do hope that in the future, many more high-quality MAGs will be available, and fast.genomics’ coverage of the diversity of bacteria and archaea will improve (see the Conclusions section).

In the revised manuscript, the section on "The fast.genomics databases" clarifies why we only include high-quality genomes. And a new paragraph in the "Limitations" section reports how many MAGs are included and discusses the trade-off between supporting presence/absence analyses and incorporating more of the sequenced diversity of bacteria and archaea.

Reviewer 2: "2) Recently Pavlopoulos et al. Nature, 2023 (Unraveling the functional dark matter through global metagenomics) published an approach that examines functional diversity beyond what was currently possible through the lens of reference genomes. Their computational approach generates reference-free protein families from the sequence space in metagenomes. They analysed over 26,000 metagenomes and identified >1 billion protein sequences with no prior similarity to any sequences from >100,000 reference genomes or the Pfam database. It would be nice for the authors to make a comparison with this approach and provide at least some measure of sensitivity with the results reported by this publication. Moreover, Pavlopoulos et al. provided a tremendous amount of novel protein families which have not been considered by the authors in their approach."

Fast.genomics does not use precomputed families, and we demonstrated that for the main database (of mostly isolate genomes), this often leads to better results. So we're not sure if it would make sense to incorporate the families identified by Pavlopoulos et al into fast.genomics.

Fast.genomics's main database does include 1,418 non-isolate genomes, so some of the families from Pavlopoulos et al do have homologs in fast.genomics already, and these can be found using fast.genomics's search tools. As metagenome assemblies improve, a greater proportion of the diversity of bacteria and archaea will be represented in fast.genomics. For now, fast.genomics's databases do not include low-quality MAGs because we want to support presence/absence analyses (see above).

---

## [Decision Letter · Decision Letter 1]

24 Mar 2024

A fast comparative genome browser for diverse bacteria and archaea

PONE-D-23-38656R1

Dear Dr. Price,

We’re pleased to inform you that your manuscript has been judged scientifically suitable for publication and will be formally accepted for publication once it meets all outstanding technical requirements.

Please, when preparing for submitting the final version of your manuscript, note the following possible typo:

- line 303: "These include fast two ways ... " should probably read "These include two fast ways ..."

Kind regards,

Vasilis J Promponas

Academic Editor

PLOS ONE

Additional Editor Comments (optional):

Reviewers' comments:

Reviewer's Responses to Questions

**Comments to the Author**

1. If the authors have adequately addressed your comments raised in a previous round of review and you feel that this manuscript is now acceptable for publication, you may indicate that here to bypass the “Comments to the Author” section, enter your conflict of interest statement in the “Confidential to Editor” section, and submit your "Accept" recommendation.

Reviewer #1: All comments have been addressed

Reviewer #2: All comments have been addressed

2. Is the manuscript technically sound, and do the data support the conclusions?

Reviewer #1: Yes

Reviewer #2: Yes

3. Has the statistical analysis been performed appropriately and rigorously? 

Reviewer #1: Yes

Reviewer #2: Yes

4. Have the authors made all data underlying the findings in their manuscript fully available?

Reviewer #1: Yes

Reviewer #2: Yes

5. Is the manuscript presented in an intelligible fashion and written in standard English?

Reviewer #1: Yes

Reviewer #2: Yes

6. Review Comments to the Author

Reviewer #1: (No Response)

Reviewer #2: (No Response)

7. PLOS authors have the option to publish the peer review history of their article (what does this mean?). If published, this will include your full peer review and any attached files.

Reviewer #1: No

Reviewer #2: No

---

## [Editor Report · Acceptance letter]

27 Mar 2024

PONE-D-23-38656R1 

PLOS ONE

Dear Dr. Price, 

I'm pleased to inform you that your manuscript has been deemed suitable for publication in PLOS ONE. Congratulations! Your manuscript is now being handed over to our production team.

Kind regards, 

on behalf of

Dr. Vasilis J Promponas 

Academic Editor

PLOS ONE